# Constraint Closure Drove Major Transitions in the Origins of Life

**DOI:** 10.3390/e23010105

**Published:** 2021-01-13

**Authors:** Niles E. Lehman, Stuart. A. Kauffman

**Affiliations:** 1Edac Research, 1879 Camino Cruz Blanca, Santa Fe, NM 87505, USA; edacediting@gmail.com; 2Institute for Systems Biology, Seattle, WA 98109, USA

**Keywords:** origins of life, constraint closure, RNA world, novelty, autocatalytic sets

## Abstract

Life is an epiphenomenon for which origins are of tremendous interest to explain. We provide a framework for doing so based on the thermodynamic concept of work cycles. These cycles can create their own closure events, and thereby provide a mechanism for engendering novelty. We note that three significant such events led to life as we know it on Earth: (1) the advent of collective autocatalytic sets (CASs) of small molecules; (2) the advent of CASs of reproducing informational polymers; and (3) the advent of CASs of polymerase replicases. Each step could occur only when the boundary conditions of the system fostered constraints that fundamentally changed the phase space. With the realization that these successive events are required for innovative forms of life, we may now be able to focus more clearly on the question of life’s abundance in the universe.

## 1. Introduction

The plausibility of life in the universe is a hotly debated topic. As of today, life is only known to exist on Earth. Thus, it may have been a singular event, an infrequent event, or a common event, albeit one that has thus far evaded our detection outside our own planet.

Herein, we will not attempt to argue for any particular frequency of life. Rather, it may be instructive to ask a related question, that of the inevitability of life, and whether that issue could be addressed by a consideration of fundamental thermodynamic concepts. In effect, it is useful to consider whether life is a deterministic outcome of matter and energy, analogous to the existence of stars. This is the question of “biocosmology.” Star formation and evolution are phenomena that originate from probabilistic events (self-gravitational compression of random, compact knots of molecular clouds) and are guided by a set of thermodynamic constraints. Life too may follow this pattern. So, what are these constraints, and how do they apply to the particularities of living systems? We will argue here that it is specifically the closure of cycles of constraints that allow for life, and that at least three such closures have resulted in life as we know it on Earth. 

## 2. Boundary Conditions 

While one can describe energy, information, etc., in terms of laws that are self-consistent and are apparently never broken, one is challenged to explain their meaning unless the boundary conditions under which they operate can be defined. Boundary conditions specify the local phase space of current and future actions, such as the barrel of a cannon or the edges of a billiard table [1,2].

As the universe grows immeasurably, epiphenomena on the scope of stars, planets, and life originate in ever-smaller corners of an ever-expanding phase space. While regions of space can exist without stars, and stars can exist without planets, so too can planets exist without life. Something must happen to change the boundary conditions of each “lower-level” phenomenon to allow for the possibility of the next “higher-level” phenomenon; the phase space must be altered in some fundamental way. 

## 3. Constraints and Work 

Boundary conditions provide constraints for a system. Constraint closure is the emergence of non-equilibrium processes with work produced as a constrained release of energy [1,2,3]. Life has been described by Pross in terms of dynamic kinetic stability (DKS): a non-equilibrium steady state in which matter and energy flow in a particular manner that allows for evolution [4,5]. A key feature of kinetics, be it in DKS or in any sort of chemical transformation, is catalysis (see below).

Critically, work is the constrained release of energy into a few degrees of freedom. The formal definition of work is the transference of energy that occurs when a force is applied to a body that is moving in such a way that the force has a component in the direction of the body’s motion. In a thermodynamic sense, the work function, ***A***, or the Hemholtz energy, is given by ***A*** = ***U*** − ***TS***, where ***U*** is the internal energy of a system. During constraint closure, the constraints actually result in the delayed production of entropy while work is being manifest [2].

The important point that we need to stress here though is that work itself can construct the constraints needed for closure [2,3]. A constraint-closed system can literally construct itself. Specifically, reproducing systems do thermodynamic work and thus create their own boundary conditions. To see how this can be, consider the following [3]. Imagine a chemical reaction in which **A** + **B** → **C**. Many such reactions would exist, even in lifeless environments. A second reaction could be imagined in which **D** + **E** → **F**. Now if the presence of the molecule **C** influences the kinetic features of the second reaction, then the two reactions are coupled. Next, imagine a third reaction **G** + **H** → **I**. Likewise, the kinetics of this reaction are influenced by the presence of molecule **F**. Now, if molecule **I** were to feed back to influence the kinetic properties of the first reaction, then a closed cycle exists. Each reaction generates work, unless the change in the internal energy ***U*** is exactly balanced by a converse change in ***TS***. However, in this example the work influences the operations of connected reactions, and the whole network of three reactions then experiences a closure that in turn creates a distinct boundary condition for the existence of the cycle itself. Here, constraint ‘closure’ means that a confluence of thermodynamic constraints has resulted in a change in the local phase space; the pathways available to the system have been canalized to such an extent that new phenomena can, perhaps counterintuitively, come into existence. It need not be coincident with a ‘closure’ of a chemical cycle, but in the example above, it is. Such a change to the phase space could not have been predicted or described beforehand [2]. Boundary conditions change as information arises and new phase space is entered. No new laws of physics need to be invoked to describe these phenomena [2].

## 4. Catalysis 

Catalysts lower the activation energy of a particular reaction to allow for an enhancement of rate constants. In doing so, they refine the boundary conditions on the release of the work from the reaction into a few degrees of freedom as reactants traverse from one energy potential into another as products, and thus also on the faster formation of a subset of the possible products. Stated more forcefully, catalysis may be the very phenomenon that established the specific boundary conditions that led to life on Earth. Catalysts can in theory exist as molecular shapes that influence the inter-conversions of other molecules, but often as condensed matter as in the surfaces of solids.

Planets can (and most do) persist indefinitely in a phase space in which catalysis exists, but not in the form of a cycle that allows for constraint closure. The tripartite cycle of chemical reactions described above is an example. Thus, a critical requirement for the emergence of life is the concomitant occupation of a locality in which chemical agents such as **A** through **I** exist, exist in a significant abundance, and exhibit the particular (and perhaps idiosyncratic) network of interconnectedness that allows for the circular topology of sequential mutual catalysis. Recent computational and experimental studies strongly support these assertions [6]. We will not address further the likelihood of this requirement in the history of the universe. This important question will be left for a future discussion, but we note that there must be a Drake Equation analog for the probabilities that each of the individual sub-requirements is met.

## 5. Autocatalytic Sets 

We have arrived at the conclusion that a kinetically constrained cycle is a requisite for life. This point has often been made before [7,8,9,10,11], but the additional point that such cycles can have arisen by, and display, constraint closure, has not. A collective autocatalytic set (CAS) is a class of cycles that fits this requirement. This is a chemical reaction network in which the molecules mutually catalyze each other’s formation from a basic food set. An autocatalytic set is an example of a Kantian Whole: in an organized being, the parts exist for and by means of the whole. A more specific type of autocatalytic set known to be relevant to life’s origins is an reflexively autocatalytic and food-generated set (RAF) [12,13]. Here, all reactions in the set are catalyzed by at least one of the molecules from the set itself and all molecules in the set can be built up from the food set through a sequence of reactions from the set itself. In considering the difference between a CAS and an RAF, it becomes clear that which molecules are included in a set—and which subsets exist within larger sets—is critical. This is particularly true of “food” and catalysts. While food can be conflated with energy, which is always needed for intermolecular conversions, in a CAS, food refers to a source of the elements and chemical moieties required for autocatalysis. The food set is a subset of the molecules that can be assumed to be available from the environment. Food molecules in an RAF are not the molecules that can participate in the inter-conversions of other molecules; they are not catalysts. Catalysts, as mentioned above, are precisely the agents that provide constraint closure of cycles leading to life. Yet another variant of an autocatalytic set is one in which at least one of the food molecules is indeed a catalyst; this is a constructively autocatalytic set, or CAF [14]. While the case may have been that the first biorelevant autocatalytic set were a CAF, we refer mainly to RAFs here.

The key point for any of these sets is closure: all molecules in the set must be produced using reactions only from the set itself. Again, we see that constraints arise from, are determined by, and are satisfied by, closure. In any type of autocatalytic set, the constraints on the future phase space are derived from the work that results from the current catalytic events; two concomitant feedback loops (an RAF and a work cycle) must operate to imbue the universe with the phenomenon of life.

So where did food and catalysts come from? The early Earth (ca. 4.5 Ga) was a vast set of molecular types. Prior to (and during) the condensation of our solar system, Miller/Urey-like processes [15], as shown recently [16], would have generated a high degree of molecular diversity. As a back-of-the-envelope example calculation, one week of such processes could generate a set of several thousand different organics [16]. Although the organics themselves could have provided some of the catalytic prowess required for closure as in a CAS, enclosure was more likely facilitated by physical entrapment and/or surface catalysis. Microenvironments such as holes in rocks [17], surfaces of drying ponds [18], protocells [19,20], or aerosol droplets [21] are intriguing candidates in this regard.

## 6. Three Major Transitions in the Origins of Life

We posit that at least three such constraint closure events have led to abiogenesis on Earth (Figure 1). Szathmáry and Maynard Smith described fundamental steps in evolutionary progress as “Major Transitions” [22]; likewise, we do so here for those involved in the chemical origins of life. 

### 6.1. Autocatalytic Sets

The first major transition **1** (Figure 1) would have been from a chaotic molecular soup to a small molecule RAF, as recently inferred from autotrophic metabolism [22], expanding on several theoretical works [7,8,10,12,13]. The chemical diversity resulting from planetary formation/evolution and the ongoing import of exogenous species from meteorites, comets, and other objects was tremendous [16]. Products of such primordial work-processes could have been sprays of molecules. 

As these molecular species got produced, eventually there would have been a phase transition into an RAF when the diversity hit a certain point. Diversity and relative abundances are the critical parameters here. A recent modeling of prebiotic chemical reaction networks demonstrated an emergence of new reaction pathways and cycles once catalysis becomes established among the molecular population [6]. It is perhaps obvious that diversity in this setting means that of chemical moieties and shapes. Miller/Urey events generate, for example, amines and carboxylic acids readily. As such reactions progress, the combination of these moieties leads to amino acids. Likewise, aldehydes, esters, and even heterocycles subsequently appear, and it is often speculated that carbon chain-length enhancement and complex sugar formation can also follow through reaction sets such as the formose reaction. Of course, this last is an autocatalytic reaction. However, it is worth pointing out that the formose reaction, or any formal RAF, derives from a random Boolean network of molecular types [23]. These nets fall in three regimes: ordered, chaotic, and in the region of a phase transition between ordered and chaotic. During the period when such nets experience low connectivity, from a chemical standpoint that means that there is high chemical specificity. In other words, molecules have attained a level of complexity in moiety and shape space such that they will only interact with certain other molecules; this is relevant in our consideration of an abiogenesis major transition when we consider them as catalysts. If there is too low of specificity, molecules can catalyze the interconversions of too many other molecules, leading to chaotic networks that contain too many connections. This is a common problem of experimental formose reactions: they engender a poorly bounded product set and do not achieve catalytic closure. However, should the set happen to produce sufficient moiety complexity and sizes, catalytic pockets arise. It has been pointed out that most current macromolecular enzymes depend on small molecules or metals for their catalytic prowess, allowing for the aforementioned reconstruction of primitive RAFs from bacterial metabolism [24]. 

When small-molecule catalysts exist in sufficient quantity and they are able to discriminate among substrates, then there is indeed catalytic specificity, and then the network connectivity can drop low enough to allow for an ordered set. Yet this is still not enough to escape triviality, i.e., futile cycles not relevant to life. The twin requirements for sufficient diversity and abundance must be met. We can at least parameterize these requisites by considering numbers and types of atoms, bonds, molecules, and reactions in a network [8,25]. As the available atomic repertoire (C, H, O, P, N, S, etc.) increases, the number of possible interatomic bonds (C–H, C–O, H–O, etc.) increases, and increases at a disproportionately faster rate with the addition of new atoms. Critically, as the ratio of available molecules to available atoms goes up, the ratio of available reactions to available molecules scales up even faster. Likewise, the ratio of network (graph) topologies to available reactions scales disproportionately, heightening the probability of autocatalytic sets until such time that they become inevitable [8]. 

With enough molecular diversity in enough abundance, and, provided catalysis came to exist to tame these explosions in a productive manner, the system itself could sufficiently affect the local phase space such that the required constraint closure became possible, and a small molecule RAF resulted. At that time then, for any planet(s) that evolved life, there arose the first simultaneous catalytic and constraint closure. 

### 6.2. Polymers

The first metabolic autocatalytic sets could have included amino acids, nucleotides, and/or similar molecules of sufficient complexity both to allow for end-to-end condensation and for information storage: the formation of “aperiodic crystals”, i.e., linear polymers. The formation of polymers from these building blocks happens sufficiently in wet/dry cycles, cf. [18]. If a metabolic RAF makes amino acids and nucleotides, any peptide or RNA (or mixture) RAF will be selected to make the same amino acids or nucleotides that are its specific food molecules. Once polymers existed, the next major transition, **2** (Figure 1) to a polymer RAF became possible. These polymers served as catalysts to further explore molecular diversity space. Specifically, the existence of polymers can allow for intermolecular recombination, the shuffling of blocks of information, as in **A•B** + **C•D** → **A•D** + **C•B**, which provides the opportunity to leap large distances in genotype and phenotype space rapidly. As shown experimentally and independently in two different laboratories, given a random pool of RNA polymers up to length *n*, recombination generates molecules of length at least up to length 2*n*−1, and possibly longer [26,27]. These recombination reactions require RNA-RNA hybridizations over regions as few as three nucleotides, allowing for the possibility of vast networks of RNA-driven RNA recombinations with relatively lenient substrate specificity requirements.

Kauffman [8] realized the benefit of recombination to facilitate greatly the closure of autocatalytic sets. Imagine a set of polymers, such as peptides or RNA, which has an a priori probability *P* of catalyzing any chemical reaction. At this point, this set is not an RAF, rather, it is merely a set of disorganized polymers that came into existence through the random condensation of monomers. Yet even if the innate catalysis in the pool allowed only for polymer length growth and shortening through joining (ligation) and cleavage (hydrolysis) reactions, respectively, autocatalytic closure would occur once polymers reached a certain length [8]. This length depends on *P*, and, for *P* values in the 10^−4^ to 10^−9^ range, the length needed to fall in the 12–27-mer range. Polymers of this length are possible from random monomer condensation, but are rare; shorter polymers would be present in much higher abundance [28]. If the catalytic capacity of the random polymers includes recombination as well, however, the necessary length range drops to 5–13, one far more abiotically obtainable. Perhaps of greater significance, one can calculate the size of the polymer set (the number of molecular species) required in order to achieve autocatalytic closure. Adding recombination to ligation/hydrolysis lowers the needed set size by four orders of magnitude, given reasonable assumptions about system behavior [8]. 

Consequently, we can speculate that polymer RAF closure—major transition **2**—occurred when the monomer condensation processes led to a large enough set of polymers, and when those molecules encountered recombination prowess, e.g., *trans*-peptidation or *trans*-esterification. In the case of RNA, both events could have happened when mere 8-mers or so became abundant [8,29]. In any event, the transition to a polymer RAF happened at the point when monomers condensed to such an extent that a threshold in the polymer length distribution was crossed, which simultaneously altered the boundary conditions for the length distribution itself and for the catalytic functionality (work) repertoire, pushing out phase space and leading to autocatalytic closure.

### 6.3. Polymerases

With an RAF of polymers, reproduction of linearly encoded information came into existence. Linear information storage simultaneously possesses the properties of minimizing Shannon entropy, maximizing channel capacity, and being chemically tractable in an RAF setting. We posit that, unlike most RNA World viewpoints of the last 50 years, that this reproduction was based on recombination rather than polymerization [25], and that it was distributed in the population in a mutualistic fashion [11]. However, recombination is inefficient at exploring local regions of phase space to discover local optima; mutations could only be of the “saltation” type envisioned by T.H. Morgan and contemporaries, cf. [30]. By contrast, an RNA replicase could exploit point mutations to fine-tune phenotypes. Moreover, under some conditions, reproduction by recombination is subject to invasion by template-directed replication, when the latter is sufficiently enabled by the environment to foster inter-self competition without falling prey to parasitism [11].

Eventually, recombination will discover polymers with slight polymerase activity, and these would be improved by selection to have replication efficiencies that allow them to fall below Eigen’s error threshold. Polymerase activity requires at least two elements that recombination does not. First, it requires that the catalytic activity of the polymer to be processive, that is, that it perform repetitive actions on one of its substrates (the template) by binding, catalyzing, releasing, and rebinding over and over again. This is a far more demanding set of actions than is required for, say, *trans*-esterification. Consequently, the secondary and tertiary structural complexity of lab-evolved replicase ribozymes, e.g., [31] matches or exceeds that of natural self-splicing (recombining) intron ribozymes despite the fact that such introns have benefited from billions of years of evolutionary refinement. Second, at least in their modern form, polymerases rely on activated substrates, those with “high-energy” bonds, such as the triphosphate in ATP. Recombination on the other hand interconverts two bonds of equivalent energy [25]. 

Once again, a constraint closure provided the wherewithal to make this major transition, **3** (Figure 1). To discover and engrain replication as the dominant mode of RNA reproduction, recombination would first swap fragments of existing RNA oligomers, creating random new combinations. Some small fraction of these would necessarily possess novel secondary and tertiary structures that could manifest catalytic functions vis-à-vis nucleic acid manipulations. Regardless of what these functions are, initially, they would alter the boundary conditions of recombination. As a simple example, if a pool of oligomers existed in which the lengths were normally distributed and centered around 20-mers, then 50-mers would be exceedingly rare. Without polymers of this length, very few complex tertiary structures (and hence complex active sites) would be possible. This then would constitute a boundary condition for the RNA population’s catalytic repertoire. Yet the recombination of two 25-mers, although uncommon, could create longer RNAs, even those in the 50-mer range. Once the length and functional boundaries have been expanded, the phase space for functionality would be altered. In the historical case of life on Earth, within this new phase space was a crude polymerase, with a low processivity and a high mutation rate. However, even such a molecule could provide small, incremental improvements to existing recombinases. This event, in turn, would spur the further exploration of deeper sequence space to discover longer and more efficient replicases. Concomitantly, the expanded phase space would allow for the discovery of catalysts that would manipulate nucleotides, for example, the creation of those with bonds of different energies such as the 5′-5′ bond, as in NppN nucleotides, proposed by Yarus to spur the evolutionary development of the RNA world [32]. Subunits of this type have the potential themselves to enhance the replication efficiencies and fidelities of polymerases. Taken together, the narrative above depicts the type of constraint closure that led to the third major transition in the origins of life. It gave rise to the type of reproduction—replication—that engendered the evolutionary path to self, cells, and life as we know it. Although our depiction centered on nucleic acids, analogous arguments could be made for peptide reproduction.

Discovery of replicase function could have happened in several specific ways; we can suggest one of them. Imagine the stacking of short RNA oligomers, each of length *n*, to form polymers of length *xn*, that fall apart to form an RNA recombinatory CAS. RNA stacking, a form of spontaneous molecular self-assembly that characterizes primitive chemical reproduction [11], can produce structures far larger than those of individual RNA strands. These structures then could explore function space more extensively, and encounter polymerase activity, as discussed above. Moreover, many such stacks would themselves have 5’ overhangs that would be natural templates for primer extension reactions, facilitating the replication of subsets of these stacked RNAs by crude polymerases. That is to say, they offload reproduction to an RAF, and gradually evolve the polymerase capable of replicating lengths in the range of its own, *xn*.

Regardless of the particulars of its history, this final major transition **3** allowed for exponential reproduction, when it could only have been sub-exponential previously. This is in contrast to the classic representation of the RNA World that posits the existence of an initial long RNA polymerase ribozyme able to copy itself; this is an unstable view. The RAF view we describe here, on the other hand, is a stable one. 

## 7. Novelty and New Niches 

In each of the transitions described above, a constraint-closed system literally constructs itself into an area of phase space that could not have been predicted a priori. In the parlance of evolutionary theory, these closures allow for the penetration of niche space into new regions previously unexplored by life. Specifically, catalytic adjustment alters the boundary conditions under which the molecules can operate, and this in turn changes the phase space of the system. The coalescence of an autocatalytic set simultaneously alters the topology of the work cycle and creates theretofore unknown epiphenomena. 

It is in fact novelty, the chances that useful novelty arises, and the rate at which this happens, that delimit the terms in the Drake equation, be it the classic one or the origins-of-life equivalent. Discovery of novelty in a rugged phenotypic landscape requires escape from local “wells” where nascent life could get trapped and succumb to entropic decay. Consider major transition **3**, above. Substantial novelty was required: here, the difficult problems of processivity and utilizing high-energy bonds (NTPs for RNA) were paramount. These required a very specific and unique manifestation of the available catalyses, not merely stepwise enhancements of existing catalysis. Novelty in catalysis pushed life into new niche spaces, which in turn further augmented the persistence of life itself. Because different causal features of the same polymers can act as different boundary conditions, the phase space created could not have been described ahead of time.

## 8. The TAP Equation and the Origins of Novelty 

Novelty can be described by a sudden expansion of the number of “species” in an environment. This phenomenon can be embodied by the TAP (theory of the adjacent possible) equation [33]:$$M_{t+1}=M_{t}+\sum_{i=1}^{M_{t}}\alpha_{i}\left(\begin{array}{c} M_{t}\\ i \end{array}\right),$$
where *M_t_* = the number of “goods” or “tools” in an economy (here: evolving system, or universe) at time *t*; the term α*_i_* is a scalar that represents the fact that it is harder to compose complex things from more items (*i*). In the case of α = 1, all possible tools can be made: the “total possible”; in the case of α < 1, only a subset of possible tools can be made: the “actual possible”. The final term in the expression, *M_t_* chose *i*, represents the number of possible combinations of *i* items that can be found in a set that is *M_t_* large. This term causes the value of *M_t_* to expand hyperbolically with time.

We can see the application of novelty to prebiotic chemistry using the relationships among atoms, molecules, bonds, reactions, and networks as discussed above. Consider the case of stepping up three stages of complexity, as intoned by Figure 1. Let us use a crude example in which *M*_1_ would be the first and most rudimentary “economy,” in which only three atoms existed (*i* = 3), say C, H, and O, in the repertoire of the primordial solar system. Allow these three atoms to combine to make four simple molecules (e.g., H_2_, CO, H_2_O, and CH_4_, although these are named arbitrarily), such that *M*_1_ = 4. Then, with α = 1, the next time step (*M*_1_ → *M*_2_) could represent the transition from molecules to uncatalyzed chemical reactions. Using the TAP equation, *M*_2_ would be 4 + _4_C_1_ + _4_C_2_ + _4_C_3_ + _4_C_4_ = 4 + 4 + 6 + 4 + 1 = 19. The next time step (*M*_2_ → *M*_3_) could represent the transition from spontaneous chemical reactions to rate-enhanced reactions upon the advent of catalysts. Now, we can begin to see the explosion in potential novelty, as *M*_3_ = 19 + _19_C_1_ + _19_C_2_ + _19_C_3_ + … + _19_C_19_ = 19 + 19 + 171 + 969 + … + 1 = 19 + 524,287 = 524,306. Further steps in complexification expand the number of “tools” in the evolutionary repertoire seemingly boundlessly. This could be true even if **α*_i_*** were to drop with *i* in a linear or even an exponential fashion; in such cases, what is created in each step is only a subset of the total possible.

From the enumeration above, three things are hopefully clear. First, novelty begets novelty, and, provided that within novelty the means to explore future phase space in an evolutionarily efficient manner exists, then the “major transitions” can become manifest events at the atomic or molecular levels during the origins of life. Second, the very means of this requisite exploration can come about through constraint closure, in which the boundary conditions of the phase space can direct canalization in a self-supporting (read, autocatalytic) fashion. Third, the TAP equation represents a general model of how pre-existing things become new types of things. We show here an example of how TAP describes the prebiotic chemical evolution in the universe that led to life through the three major transitions of Figure 1, but it would be equally suitable to describe the evolution of cells and of higher biological structures.

## 9. Discussion

We have provided insights into how nascent life overcame three significant hurdles to arrive at the indefatigable force that it became on the early Earth. These challenges were met by work constraint cycles, which generate closure events at the same time they spawned novelty. During life’s origins, necessarily a set of chemical events, these cycles must have been chemical ones. With each closure, the complexity of the system jumped, creating new epiphenomena, and ultimately made the chemistry-to-biology transition. Originally, persistence is all that was selected for, as in the DKS paradigm. As Dennett put it [34], “Before we can have competent reproducers, we have to have competent persisters, structures with enough stability to hand around long enough to pick up revisions.” However, persistence was eventually superseded by innovative reproduction.

We wish to make clear that nascent life must have operated on principles that may be hard to recognize in contemporary life. Specifically, the first stages of life were shaped in large part by the phenomena that allowed abiogenesis from chemical soups in the first place. We draw on a large body of literature that strongly suggests that CASs were the conduit from chaos to order [1,8,9,11,13,14,23]. Autocatalysis provides the means for the exceedingly rare phase changes required to step up to fundamentally higher levels of complexity. In modern life, by contrast, other mechanisms for reproduction, such as polymerases and ribosomes with discrete sequences but broad substrate specificity, have come to dominate. Nonetheless, there are relics of CAS existence in primary metabolism today [24]. It is simply no longer reasonable to search for the mechanisms of life’s origins motivated by the same phenomena we see at the large molecule or cellular levels.

Herein, we attempt to advance our understanding of the origins of life by breaking it into three distinct stages, and by drawing parallels between chemical closures (CASs) and work closures (constraint closures). Both must have occurred, and at the same time: the first to allow life to get a foothold in an otherwise purely chemical setting, and the second to transit from one realm of complexity to another. Note that CASs do not appear and evolve magically. They are driven by energy flow and by entropy attenuation, the former provided by sunlight and chemical bond rearrangement, and the latter by space partitioning through encapsulation, for example. Their change over time required diversity and selection, as did all biotic systems [9].

It is illustrative to realize that one could not have predicted the exact sequence of events that led to life, even if one has the conviction that life is inevitable in the universe. Novelty is inherently unpredictable. Ask, for example, how many uses are there for a screwdriver? The answer is neither a finite nor an infinite number. Rather, it is an indefinite one. The tool was invented to solve a specific problem, but quickly found myriad other uses. A strong, flat piece of metal is adept at tightening screws with a matching slot, and by chance can manipulate many other objects, but only those that can also accept the shape of this metal. The constraints provided by functions that exist at any point in time create constraints for future functions; in biological evolution, there is no deduction, only induction, and the phenomenon persists without invoking any new fundamental laws. By this analogy, we see that living systems can create their own phase space. Life as we know it on Earth required all three major transitions described above. This clearly happened on the Earth, but on other locations perhaps only the first or second transition(s) occurred. Mars or Enceladus could in fact be such cases.

Yet on Earth, primitive life indeed made the saltation into that region of phenotypic phase space in which nucleic-acid replicases—polymerases—provided the wherewithal to search efficiently the landscape for better and better selfish entities capable of eluding the catastrophes posed by both the error threshold and parasites. The co-evolution of polymers into a polymerase may be testable now in the laboratory, and this should be a major thrust of experimental efforts in the field. Once a polymerase activity exists, then the principles discussed by Eigen [35] can take place in terms of quasispecies. Later, coding became embodied in the system, but this is beyond the scope of the current discussion. Once rudimentary life itself existed, then larger-scale major transitions led to complex life [22].

## 10. Conclusions

Life may be common in the universe, or quite rare. To reach this level of complexity, a chemical system must have met by chance a series of requirements, including diversity, abundance, catalytic prowess, and the capacity to form self-closing cycles: collective autocatalytic sets. We are pointing out here that these requirements were obligatory interdependent via work cycles. In particular, the origin of catalysis went hand-in-hand with the origin of CASs. Catalysis acts to form smaller subsets of all possible things, generating constraints. When these are bounded by closure events, perhaps counter-intuitively, novelty is spawned. Three fundamental innovations, or major transitions, led to life as we know it on Earth today. Such constraints at first were solely of a chemical kinetic nature, but later they came also to be structures: cells can construct their own boundary conditions and potentiate constraint closures [2].

## Figures and Tables

**Figure 1 entropy-23-00105-f001:**
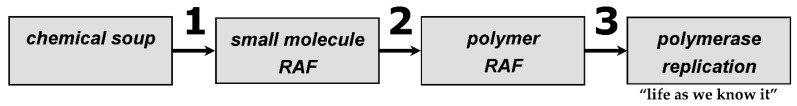
The three major transitions through constraint closure during the origins of life on Earth. (**1**) The advent of a reflexively autocatalytic and food-generated (RAF) set that arises from a diverse population of organic molecules upon autocatalytic closure. (**2**) The advent of polymeric information, also an RAF set, from a subset of existing RAF sets. These mutualistic [11] polymers reproduce and explore niche space via recombination of large blocks of information. (**3**) The discovery of polymer replication (unit-by-unit) reproduction and the resultant “selfish” genotypes and their inter-self competition [11].

## Data Availability

Data sharing not applicable.

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
