# Peer review of "Constraint Closure Drove Major Transitions in the Origins of Life"

_entropy, 2021, doi:10.3390/e23010105_

Round 1

Reviewer 1 Report

Constraint closure – Lehman and Kauffman

This paper discusses the roles that collective autocatalytic sets may have had in the origin of life. It discusses three stages separately – small molecules, informational polymers, and polymerase replicases. There are a lot of ideas here but I am still a long way from agreeing with (or understanding) everything that is in this paper.

line 65: 'the first reaction'. Shouldn't this be 'the second reaction'?

The words 'constraints' and 'closure' are used very generally at the beginning. It is not very clear exactly which phenomena might be included in these words. The example given around line 70 is clear in that the three reactions form a closed cycle. The rest of the paper is indeed talking about closed reaction cycles, so maybe it would be clearer to simply say this at the beginning, without introducing very broad discussions of phase space, boundary conditions, stars etc, which is a bit distracting.

Lines 94-96: I have to be honest and say that I have not understood in what sense a 'kinetically constrained cycle' is a requisite, and why the idea of constraint closure discussed here is different from what has been shown before.

I understand that this paper is saying that (i) we need to make monomers, then (ii) we need to make polymers, then (iii) we need to replicate the polymers. I would agree with this, and I agree that clearly separating these steps is useful. This paper is written in an RNA-World context (or at least it discusses replication of informational polymers), which is helpful, because early papers of Kauffman on autocatalytic sets were very vague about what kind of molecules were being discussed, and it is often considered that the autocatalytic set theory is somehow an alternative to replication-first theories like the RNA World. This paper goes some way towards linking metabolism-first and replication-first ideas and saying why they are both needed.

However, I am not convinced that autocatalytic sets are necessary for step (i). If step (iii) is RNA replication, then what the small molecules have to do at step (i) is generate lots of single nucleotides. But why do the reactions that generate nucleotides need to be autocatalytic? There needs to be an environment with some kind of free energy source – e.g a source of small molecules that are out of equilibrium that are assumed to be present with an essentially inexhaustible supply. These molecules can drive a set of reactions that (hopefully) will produce nucleotides at a constant rate. It is not clear that any of the reaction steps need to be catalyzed by any of the small molecules present. It would be simpler to get started if they were spontaneous uncatalyzed reactions. There has been a lot of previous work on the probability of closed autocatalytic sets being present in reaction networks, but I just don't see why this is a requirement. Is there any evidence that small molecules do actually catalyze the kinds of reaction steps that would be involved in synthesis of nucleotides? If some of the steps do happen to be catalyzed, why would all of them have to be? Why would it be a closed set? And if nucleotides are the essential product that we want to generate, then is there any evidence that nucleotides can catalyze any of the previous steps involved in their synthesis? If not, then the nucleotides would not be involved in the CAS, even if there were one. To insist that a CAS is required at the small-molecule stage seems to be a red herring.

If some of the reactions involving small molecules were catalyzed, this might be beneficial, because they would go faster and might speed up the process of creation of nucleotides. But this could come later via evolution. In a cell we have lots of small molecules involved in metabolism, and before we can divide a cell we have to double the quantities of all the molecules. So I guess it makes sense to say that these molecules are all part of an autocatalytic set that creates more of itself. But the catalysts are enzymes, not small molecules. We might imagine that proteins or ribozymes later evolved to catalyze the small-molecule reaction steps that were already occurring. But this can only occur after steps (ii) and (iii).

The discussion of step (ii) considers ligation and cleavage reactions (possibly driven by wet/dry cycles) plus recombination. It seems likely that a self-sustaining mixture of oligomers could be generated by these reactions. The paper does not mention non-enzymatic primer extension or template-driven ligation. These would also be possible in such a mixture, and may have been more important than recombination. It seems odd to discuss recombination without templating.

The paper needs to be more specific on the details of the recombination reactions that are envisaged. There are several possibilities. Is it imagined that recombination is a relatively easy and non-specific reaction? Maybe all sequences (or at least a substantial fraction of random sequences) undergo recombination at some average rate? Or is it imagined that there is a specific recombinase with a well-defined sequence (like the Azoarcus recombinase) that would be a very rare occurrence in a mixture of sequences? If it is a specific sequence, is its substrate also specific – i.e. it needs a sequence that matches it in some way in order to work? If so, are we sure that the right substrate sequence would actually be present in the mixture? I am not convinced that a specific recombinase would ever create more of itself. How would there ever be a second copy? Maybe you want to argue that there is a whole CAS of specific recombinases that all just happen to assemble one another? I do not find this plausible. If there were a mixture of oligomers with ligation and cleavage and random recombination, I think it would have almost no sequence information and you would never get the same sequence twice. Maybe there was really a stage where it was like this, but I can't see how a specific recombinase would emerge and reach high frequency in this mixture. In summary, for step (ii), I can envisage a self-sustaining set of more-or-less random oligomers, but I don't see how there could have been a specific closed autocatalytic set in such a mixture, and I don't see why it was a requirement that there should have been a CAS in order to get from step (ii) to step (iii).

Line 238 – 'Eventually recombination will discover polymers with slight polymerase activity'. Yes, but the issue is how this sequence will feed back to generate more copies of itself. There would need to be a complementary sequence to the polymerase in the mixture. If the mixture is only undergoing recombination, I can't see how this would happen. If there were non-enzymatic templating too, it is slightly more hopeful. But the error rates of non-enzymatic primer extension are very large, so we have the error threshold problem, and we are limited to very short sequences. Even if there were non-enzymatic templating in the mixture as well, it may be that a sequence would have to detach and rebind to several templates over several cycles, so it would not be the complement of any one specific sequence. So there would still be very little sequence information in a mixture that was undergoing non-enzymatic templating. (Nevertheless, we are here, so something like this must have happened.) The paragraph beginning line 250 says something about how replication could emerge in the mixture, but it seems very incomplete. It is argued that recombination sometimes generates a 50mer that would be rare if there were just ligation and cleavage. OK, but how do you maintain and replicate the polymerase once you have generated the first copy by random recombination? The part about 'expanding the phase space' does not answer this question.

Line 271 says that constraint closure led to this third transition (i.e. the creation of polymerases). I don't know what constraint closure means in this case. It does not seem to mean closure of an autocatalytic set at this point, but what does it mean?

I do not see how CASs are relevant at this stage. It seems significant to me that the central processes of DNA and RNA replication and protein translation in modern cells do not work via CASs. There is a single kind of DNA polymerase that copies all DNA independent of its sequence, not many kinds of sequence-specific polymerases that only copy a little bit of DNA. Similarly, there is one kind of ribosome that makes all the proteins independent of their sequence. In order to get the RNA World going, we need a single kind of RNA polymerase that copies all RNAs independent of their sequence. Once we have that, we are free to evolve many other sequences with other functions, all of which can be copied by the same polymerase. Such a polymerase would not be part of a CAS. So my bottom line is that I broadly agree with the three steps in this paper, but I am not convinced that CASs are required at any of them.

Author Response

Dear Editor,

We thank you and three high-quality reviewers for making suggestions on our manuscript, 'Constraint closure drove major transitions in the origins of life.' Two of these reviews (#2 and #3) are enthusiastically favorable, and recommend immediate publication pending a few minor (and salient) suggestions for clarification. The third reviewer (#1) 'broadly agrees with the three steps in this paper but is skeptical of the role that collective autocatalytic sets (CASs) play in these steps. This reviewer is nevertheless pleased that this paper can help bring a concrete RNA-based exemplification to Kauffman's theories regarding autocatalytic sets. S/he makes many other excellent points, and would like a large amount of additional explanations. While we have addressed some of these in our revision, the manuscript is very long as it is, and many of these issues were clarified in detail in the publications we cite. In particular, the topics, and relevant references that provide abundant background on them, are the following:
*explanation of CAS logic and function: [7-10, 12-14]
*CAS in primary metabolism: [23]
*mechanism and specificity of recombination: [25-27]
*constraint closure: [1,3]
*small-molecule catalysis and autocatalytic action: [6]
  The alterations we have made are in red type in the revision, with page numbers referring to the original submission. These changes are summarized as follows:
1. Line 5: corrected email address for first author.
2. Line 52 (in response to reviewer #2): 'infra vida' has been replaced with its English meaning, 'see below.'
3. Line 65 (in response to reviewer #1): 'first' changed to 'second'.
4. Lines 71-72 (in response to reviewer #1): sentence changed to explain better what we mean by 'constraint closure.'
5. Line 80 (in response to reviewer #2): 'is' changed to 'may be.'
6. Line 105 (in response to reviewer #3): a sentence has been added to discuss the distinction between food and energy.
7. Line 201 (in response to reviewer #1): a sentence has been added to discuss the specificity requirements of recombination.
8. Lines 277-279 (in response to reviewers #1 and #3): two sentences have been replaced with three sentences that better explain the roles of self-assembly, primer extension, and the recombination-to-polymerase transition.
9. Line 350 (in response to reviewers #1 and #3): two new paragraphs have been added to discuss better the critical role of CASs, and other points.
10. Line 355 (in response to reviewer #2): a new sentence has been added to explain better the screwdriver analogy.
11. Line 401 (in response to reviewer #2): page numbers for the Wolos (2020) paper have been added.
12. Lines 421-422 (in response to reviewer #3): the Damer & Deamer reference has been updated.

Reviewer 2 Report

After a thorough review of the manuscript, "Constraint closure drove major transitions in the origins of life," I can see a fascinating hypothesis about the major steps in the origin of life. It is a very well written paper with a perspective based on the thermodynamic concept of work cycles. As with all the hypotheses and proposes in the field, the manuscript is not free of speculations and weak statements. However, the text's central idea could be important for discussion on the area, and the academic level is outstanding. So, in my opinion, the manuscript could be accepted in the present form.

Minor concerns and suggestions:
Page 2, line 52: "Infra vida" is a concept not free of teleological bias. Is it necessary?
Page 2, line 80: It is an essential point of the work. However, it is an asseveration that is impossible to test, so I suggest modifying the sentence as a suggestion.
Page 8, lines 355-358: A broader development of this sentence would be pertinent.
There are some errors or gaps in the references (e. g., reference 6)

Author Response

We thank you and three high-quality reviewers for making suggestions on our manuscript, 'Constraint closure drove major transitions in the origins of life.' Two of these reviews (#2 and #3) are enthusiastically favorable, and recommend immediate publication pending a few minor (and salient) suggestions for clarification. The third reviewer (#1) 'broadly agrees with the three steps in this paper but is skeptical of the role that collective autocatalytic sets (CASs) play in these steps. This reviewer is nevertheless pleased that this paper can help bring a concrete RNA-based exemplification to Kauffman's theories regarding autocatalytic sets. S/he makes many other excellent points, and would like a large amount of additional explanations. While we have addressed some of these in our revision, the manuscript is very long as it is, and many of these issues were clarified in detail in the publications we cite. In particular, the topics, and relevant references that provide abundant background on them, are the following: *explanation of CAS logic and function: [7-10, 12-14] *CAS in primary metabolism: [23] *mechanism and specificity of recombination: [25-27] *constraint closure: [1,3] *small-molecule catalysis and autocatalytic action: [6] The alterations we have made are in red type in the revision, with page numbers referring to the original submission. These changes are summarized as follows: 1. Line 5: corrected email address for first author. 2. Line 52 (in response to reviewer #2): 'infra vida' has been replaced with its English meaning, 'see below.' 3. Line 65 (in response to reviewer #1): 'first' changed to 'second'. 4. Lines 71-72 (in response to reviewer #1): sentence changed to explain better what we mean by 'constraint closure.' 5. Line 80 (in response to reviewer #2): 'is' changed to 'may be.' 6. Line 105 (in response to reviewer #3): a sentence has been added to discuss the distinction between food and energy. 7. Line 201 (in response to reviewer #1): a sentence has been added to discuss the specificity requirements of recombination. 8. Lines 277-279 (in response to reviewers #1 and #3): two sentences have been replaced with three sentences that better explain the roles of self-assembly, primer extension, and the recombination-to-polymerase transition. 9. Line 350 (in response to reviewers #1 and #3): two new paragraphs have been added to discuss better the critical role of CASs, and other points. 10. Line 355 (in response to reviewer #2): a new sentence has been added to explain better the screwdriver analogy. 11. Line 401 (in response to reviewer #2): page numbers for the Wolos (2020) paper have been added. 12. Lines 421-422 (in response to reviewer #3): the Damer & Deamer reference has been updated.

Reviewer 3 Report

This is a thoughtful, clearly written manuscript that summarizes ideas developed by the two authors over years of work. It could be published immediately, but the authors may wish to consider some points that came to mind as I read.

Energy is introduced in section 3, but then it gets lost in section 5 where it is mixed up with the word "food". For non-photosynthetic life today, food is both a source of nutrients AND a source of chemical energy. This should be clarified for readers who might otherwise be confused.

The authors should also make clear that CAS do not function magically. Instead the catalyzed reactions are driven by a constant input of energy such as the sunlight and chemical energy that are essential for all life today. I'm sure they authors understand this but they should make sure their readers do.

The authors should also point out the fundamental process that helps a "chaotic molecular soup" take the first step toward order for free, that is, without a direct input of energy. That process is called self-assembly and is as essential for the origin of life as it is for today's life. (The authors cite Eigen in this regard, ref. 35.) It is not a chemical process, but instead falls into the realm biophysics. There are three obvious examples of self-assembly. The first is the way that amphiphilic molecules become spontaneously organized into membranous compartments that can encapsulate the polymeric systems of catalysts and informational polymers. The second is the assembly of nucleotides in nucleic acids into double helical structures required for replication, storage and transfer of genetic information, and the third is the folding of polymers into catalytically active structures. The authors know all this, but their readers might not understand that in the absence of self-assembly, life would not be possible. In fact, I would rank it among the constraints that the authors are considering.

Finally, the authors should point out that for CAS to evolve, they must be capable of undergoing selection. The most efficient way for selection to occur is that a variety of systems are present within populations and then subjected to various environmental stresses. But how did populations of systems emerge on the prebiotic Earth? The simplest is that they become encapsulated in microscopic compartments, each different from all the rest. Most of the compartments are inert and their components are recycled, but a few are better able to withstand those stresses and have the quality of persistence. They might happen to contain components capable of functioning as CAS that add robustness in some way.

Reference 18 cites a 2015 publication. A more recent and expanded reference is:

Damer B, Deamer D. The hot spring hypothesis for an origin of life (2020). Astrobiology 20:429 - 452.

Author Response

Thank you 
